# Encapsulation with Natural Polymers to Improve the Properties of Biostimulants in Agriculture

**DOI:** 10.3390/plants12010055

**Published:** 2022-12-22

**Authors:** David Jíménez-Arias, Sarai Morales-Sierra, Patrícia Silva, Henrique Carrêlo, Adriana Gonçalves, José Filipe Teixeira Ganança, Nuno Nunes, Carla S. S. Gouveia, Sónia Alves, João Paulo Borges, Miguel Â. A. Pinheiro de Carvalho

**Affiliations:** 1ISOPlexis, Center for Sustainable Agriculture and Food Technology, University of Madeira, Campus Universitário da Penteada, 9020-105 Funchal, Portugal; 2Grupo de Biología Vegetal Aplicada, Departamento de Botánica, Ecología y Fisiología Vegetal-Facultad de Farmacia, Universidad de La Laguna, Avenida, Astrofísico Francisco Sánchez s/n, 38071 La Laguna, Spain; 3Faculty of Exact Sciences and Engineering, University of Madeira, 9020-105 Funchal, Portugal; 4CENIMAT|i3N, Department of Materials Science, School of Science and Technology, NOVA University Lisbon and CEMOP/UNINOVA, 2829-516 Caparica, Portugal; 5CiTAB, Centre for the Research and Technology of Agro-Environmental and Biological Sciences, University of Trás-os-Montes and Alto Douro, Quinta de Prados, 5000-801 Vila Real, Portugal; 6Faculty of Life Sciences, University of Madeira, Campus Universitário da Penteada, 9020-105 Funchal, Portugal

**Keywords:** agriculture, biostimulants, encapsulation, chitosan, alginate, guar gum, xanthan gum, carrageenans, natural biopolymers

## Abstract

Encapsulation in agriculture today is practically focused on agrochemicals such as pesticides, herbicides, fungicides, or fertilizers to enhance the protective or nutritive aspects of the entrapped active ingredients. However, one of the most promising and environmentally friendly technologies, biostimulants, is hardly explored in this field. Encapsulation of biostimulants could indeed be an excellent means of counteracting the problems posed by their nature: they are easily biodegradable, and most of them run off through the soil, losing most of the compounds, thus becoming inaccessible to plants. In this respect, encapsulation seems to be a practical and profitable way to increase the stability and durability of biostimulants under field conditions. This review paper aims to provide researchers working on plant biostimulants with a quick overview of how to get started with encapsulation. Here we describe different techniques and offer protocols and suggestions for introduction to polymer science to improve the properties of biostimulants for future agricultural applications.

## 1. Introduction

The United Nations has set 17 goals for global sustainable development. The second goal is zero hunger by 2030. To achieve this, agricultural production must be doubled, but some studies indicate that yield developments are currently not sufficient to reach this goal [1]. One of the different reasons for this is climate change. Heat waves, heavy rainfall, and rising sea levels are predicted for the 21st century, with drought, floods, and salinization among the most critical consequences for food production [2]. Today, abiotic stress in plants is the principal cause of severe yield losses of 50–80%, depending on the crop and geographical location [3]. This is a primordial problem to solve in the upcoming years. This alarming situation is an excellent opportunity for plant scientists to apply their knowledge in an interdisciplinary way in agriculture to increase productivity under abiotic stress [4].

Biostimulants (Bs) are understood as “A product that stimulates plant nutrition processes independently of the product’s nutrient content, with the sole aim of improving one or more of the following characteristics of the plant or the plant rhizosphere: (a) nutrient use efficiency; (b) tolerance to abiotic stress; (c) quality traits; or (d) availability of confined nutrients in the soil or rhizosphere are probably one of the most intelligent agricultural technologies to counter abiotic stress” [4,5]. Biostimulants are extensively covered in plant science research, highlighting that they are an environmentally friendly way to manage biotic and abiotic stresses and increase food production [6,7,8,9,10]. However, one of the cornerstones of Bs is their ease of degradation in the field, which prevents bioaccumulation. Unfortunately, this means that they must be administered continuously [11], which affects profitability in the field [12]. Furthermore, > 90% of agrochemicals expire during application [13]. Therefore, new approaches to enhance the shelf life of biostimulants are the key to improving treatment under field conditions.

Encapsulation science applied to agriculture attempts to solve the above problems that Bs have when applied directly [14]. The advantage of this active ingredient encapsulation is the slower release of the encapsulated ingredients and their more efficient use, as well as better protection (safety) for users and the environment [15]. The search for new carriers for protection in agriculture has come a long way, mostly based on the protection of beneficial microorganisms from environmental degradation [16]. However, the encapsulation of active compounds in agriculture is practically focused on pesticides [17] and fertilizers [18]. However, Bs encapsulation studies are scarce, even though they are an interesting and urgent field to develop in agriculture [19].

This review paper aims to help beginners who want to get into the science of encapsulation to make eco-friendly capsules using natural polymers. We also try to present some techniques that can be easily used by newcomers. Finally, the economic viability and applicability of a future formulation to be used under field conditions are discussed.

## 2. Encapsulation Techniques

### 2.1. Encapsulation Using Electrospray Techniques

Electrospray is a technique derived from electrospinning. Electrospinning is a physical process in which polymer fibers are drawn by the action of an electric field without the need to apply any other additional mechanical energy [20]. In electrospinning, the polymer is held in a syringe (molten or in solution) connected to a positively charged metal tip with the desired diameter, and a collector that serves as an anode where the fibers are deposited. When the electrostatic forces overcome the surface tension of the liquid, a polymer jet is formed, which is drawn from the Taylor cone to the collector [21]. Solution properties, such as concentration, solvent, and viscosity, as well as process parameters, such as flow rate, distance from the needle tip to the collector, and tension, dictate the morphology of the fibers or the formation of a spray. Electrospray occurs when the viscosity of the solution is low and does not allow the formation of a polymeric fiber. In this case, the jet emerging from the Taylor cone breaks down into droplets from which particles of different sizes and shapes are formed (Figure 1) [22]. Micro- and nanoparticles obtained by electrospray can have higher charge efficiency and narrower particle size distribution compared to particles obtained by other techniques [22]. At the writing time of this review, few encapsulation reports using electrospray were found for agriculture applications. Zhang et al. [23] performed a pesticide encapsulation using microparticles made by the electrospray technique. Zhao et al. [24] provided an interesting and explained protocol using alginate beads and CaCO_3_ with the experimental set-up.

### 2.2. Encapsulation Using Ionotropic Gelation Techniques

Ionotropic gelation is a technique that allows the production of micro- and nanoparticles by electrostatic interactions between two ionic species under certain conditions. At least one of the species must be a polymer. When a drug or bioactive molecule is added to the reaction, it can become trapped between the polymer chains, causing it to become contained within the microparticle/nanoparticle structure [25]. Hydrogel beads are prepared by dropping a drug-loaded polymer solution into an aqueous solution of multivalent cations (Figure 2). The cations/anions diffuse into the drug-loaded polymer droplets and form a three-dimensional network of ionically cross-linked units [26]. The concentration of the polymer and cross-linking electrolyte, solution flow velocity, temperature, pH of the cross-linking solution, and drug concentration are the key parameters for the particle size [27]. Ionotropic gelation has enormous advantages. The technique is relatively cheap, fast, and does not require expensive equipment or reagents. It has high encapsulation efficiency (close to 100%) when the polymer and drug concentration are optimal, and the use of biodegradable and biocompatible polymers provides formulations with mucoadhesion and other relevant biological properties [25]. The ionotropic structure is the simplest compared to electrospray techniques, but the particle size is very large compared to the aforementioned techniques [28]. Ionotropic gelation techniques have been used in agriculture to encapsulate Bs such as silicon [29] and gibberellic acid [30], with significant improvements in the behavior of Bs. An example of this encapsulation method is provided by Koukaras et al. [31], who presented us with an interesting and explained protocol using the ionotropic gelation technique with chitosan and tripolyphosphate.

### 2.3. Encapsulation Using Emulsion Techniques

This technique consists of the formation of spherical droplets within a continuous phase. The small droplets can be created with the application of a mechanical shear force. The droplets are hardened either using chemical or physical cross-linking. Oil-in-water emulsions (O/W) (Figure 3a) allow the encapsulation of hydrophobic drugs. The polymer is dissolved in an organic solvent with the hydrophobic cargo, and then the solution is dispersed in water, with the aid of an emulsifier agent [32,33] and shear forces. The hardening of particles is achieved by solvent evaporation. The bioactive compound must be compatible with the organic solvent. The water-in-oil technique (W/O) (Figure 3b) is the inverse of this technique. In this case, a polymer is dissolved in water and dispersed in oil [34]. It is possible to add different layers of water and oil phases. Water in oil in water (W/O/W) is a technique that allows for better encapsulation of hydrophilic drugs. This double-emulsion method consists of the first water-in-oil emulsion being added to a water bath [35]. Double-walled microspheres can be achieved by the water in oil in oil in water (W/O/O/W), allowing for better protection of the drug, a lower initial burst, and a more prolonged release [36,37].

Another type of production is Pickering emulsions (Figure 3c), where particles self-assemble with solid particles at the water/oil interface [38]. The particles give stability and functionality to the emulsions [39]. Yaakov et al. [40] used the Pickering emulsion formulation to encapsulate *Bacillus thuringiensis serovaraizawai* (BtA) for future pest control applications, using silica nanoparticles. In another study, zinc oxide nanoparticles were used in Pickering emulsions for pest control [41].

The use of emulsions can produce micro- and nanosized particles. This will depend on different factors such as viscosity, time, droplet formation method, and frequency, among others. Both hydrophobic and hydrophilic drugs can be encapsulated with this method or loaded afterward. However, a drawback is the necessity of cleaning the oil phase and surfactants, where centrifugation, multiple washings, filtrations, and the use of ethanol can be used [33,36,42,43,44]. Cleaning procedures should not affect encapsulated cargo nor change the sphere’s properties. The use of organic solvents and chemical crosslinkers, such as glutaraldehyde, is a concern due to environmental and toxicity problems. Chemical crosslinkers can be changed to physical ones, or chemical blockers, such as glycine for glutaraldehyde, are also viable [45,46,47,48]. Another setback is that mechanical forces may affect the bioactive agent [49].

## 3. Encapsulation Polymers

### 3.1. Alginate

Alginate obtained from different species of brown macroalgae contains different proportions and sequences of β-(1→4)-linked D-mannuronic acid residues (M) and α-(1→4)-linked L-guluronic acid residues (G) residues, which determine the alginate molecular weight and physical properties, even for the structures obtained from it [50]. Commercially alginates are available in sodium, potassium, or ammonium salts, with molecular weights ranging from 60 to 700 kilodaltons [51]. The alginate polymer consists of two monomer units of M and G (Figure 4A). Alginate’s basic structure consists of linear, unbranched polymeric units composed of monomers arranged in blocks with interspersed M and G residues, with regions containing an alternating M-G sequence within the structure (Figure 4B) [51,52].

One of the most valued properties of alginate application in the food industry is its ability to form an ionic gel in the presence of multivalent cations (ionotropic gelation). In addition, compared to other polysaccharides, alginate can form a gel regardless of temperature [53]. An explained protocol to form alginate microcapsules is provided here [54]. The gel formed by this interaction is widely used for the encapsulation of bioactive compounds in food, pharmaceutical, and biotechnological industries [55]. It can be prepared in the form of micro- and nanocapsules, with more common use for the first preparation [56]. The binding of divalent cations to alginate is a highly selective process, and the affinity of alginate for the cations increases in the order of Mn < Zn, Ni, Co < Fe < Ca < Sr < Ba < Cd < Cu < Pb [55]. The different cross-linking ions affect the ionotropic gelated alginate structure differently in comparison with Ca^2+^.Ba^2+^, making it stronger with smaller size and stable microcapsules in acidic and neutral pH environments [57]. Sr^2+^ increased the release speed and was found to be suitable for living cell entrapment [58]. Al^3+^ affected the morphology and release profile of beads [57]. Sr^2+^ increased the release speed and was found to be suitable for living cell entrapment [58]. Al^3+^ affected the morphology and release profile of beads [59], Fe^3+^ was useful to obtain prolonged release profiles [60], and Zn^2+^ had slower release profiles when compared to beads gelated with calcium [61]. Alginate has some interesting properties for biostimulants delivery for agricultural purposes [42,62]: (i) it is readily available and relatively inexpensive; (ii) it contains ingredients that are recognized as food additives; (iii) the polymer is biocompatible and does not accumulate in any organ of the human body; (iv) it is biodegradable; (v) alginate is water soluble, so environmental problems associated with solvents can be minimized; (vi) it forms a gel at room temperature, reducing the risk of the degradation of thermosensitive compounds.

Encapsulation with alginate is used in some agrochemical formulations, especially to control the release of active ingredients [63], for example, to control the release of pesticides [64,65], herbicides [66], or fungicides [67]. Another use is the optimization of the fertilization process by encapsulating various fertilizers such as biochar [68] or biofertilizers [69]. In the agricultural industry, alginate is also used as a superabsorbent polymer to coat seeds, fruits, and vegetables and as a carrier of bacteria and fungi to promote plant growth and biocontrol [70]. Alginate hydrogel combined with some amino acids has been used as a coating agent, to be used in seeds for commercial applications, to stimulate the early growth of plants and thus leading to higher crop yields [71]. Alginate beads have been used to improve the properties of biostimulants, especially in microorganism-based formulations [72]. Encapsulation within alginic matrixes attempts to increase the viability of cells and use them for various purposes, such as increasing the tolerance of plants to drought [73,74,75] or salinity [76,77,78] and the uptake of nutrients [79]. Encapsulation of microorganism-based biostimulants with alginate is a promising way to extend their shelf life [80]. Alginate beads can improve the properties of biostimulants, are capable of increasing plant growth of lettuce [81], and can increase wheat yield under drought conditions by entrapping N_2_-fixing bacteria [82]. However, at the time of writing, the properties resulting from encapsulation have hardly been researched under field conditions, as there are no corresponding production trials. In addition, we could not find any use of alginate for encapsulating other types of biostimulants as pure or organic agents against abiotic stress. Alginate in our opinion can be an interesting and profitable way to improve plant biostimulant dosage in field conditions, as is demonstrated in other interesting agrochemical products [83]; however, further research is needed.

### 3.2. Chitosan

Chitin is a linear polysaccharide composed of N-acetyl-D-glucosamine units linked by β-(1,4) (Figure 5) and is considered one of the substances with the highest production rate and biodegradability in nature [84]. Chitin is a very common biopolymer found in the exoskeleton of crustaceans, in the cuticle of insects, in algae, and in the cell wall of fungi [85]. Nowadays, chitin extraction on an industrial scale is performed from marine shellfish waste and is carried out using chemical methods. The methodology consists of three main steps: deproteinization with the addition of an alkaline solution, demineralization with an acidic solution, and finally decolorization with an alkaline solution [86]. Despite the exceptional properties of chitin, its use is very limited due to its poor solubility [87]. For this reason, the polymer industry is turning its attention to chitosan.

Chitosan is a natural and hydrophilic polymer. It is a deacetylated derivative of chitin. When the degree of deacetylation of chitin reaches about 50%, depending on the origin of the polymer, the chitin becomes soluble in aqueous acidic media and is called chitosan [88]. This solubilization occurs by protonation of the -NH2 function at the C-2 position of the repeating D-glucosamine unit. The polysaccharide is converted into a polyelectrolyte in acidic media [89]. The solubility of the polymer depends on the degree of acetylation and the molecular weight [90]. Chitosan oligomers are soluble in a wide pH range, from acidic to basic. In contrast, chitosan samples with higher molecular weight are only soluble in acidic media, even at high degrees of deacetylation [90]. Chitosan is found in some fungi, such as the Mucoraceae family [91]. Traditionally, commercial chitosan samples have been produced mainly by chemical deacetylation of chitin from crustaceans, but recently, chitosan from fungi [92] and insect cuticles [93] has gained interest. Polymer properties are tightly related to the physicochemical properties of the chitosan (mainly molecular weight and acetylation degree). Therefore, when working with chitosan to produce reproducible results, a good and completed polymer characterization is mandatory [90].

Chitosan is typically used to develop nanoparticles [94], being an ideal option as a carrier due to its biocompatible and biodegradable properties [95]. In addition, chitosan is particularly interesting to be used in agriculture, as it can be absorbed by plant surfaces (e.g., leaves and stems), increasing the contact time between enclosed substances and plant tissues [96]. In addition, chitosan nanoparticles facilitate the transfer through the cell membrane [97]. The most used method for producing chitosan nanoparticles is ionotropic gelation, which is achieved by the inter- and intramolecular cross-linking of the polycationic chitosan by an anionic cross-linking agent, such as the most used tripolyphosphate [98]. Chitosan is dissolved in an aqueous acetic acid solution, and the aqueous solution of tripolyphosphate is added dropwise to the chitosan solution. Finally, nanoparticles can be formed immediately under mechanical stirring at room temperature [99].

Chitosans are widely studied as plant biostimulants because of their properties against abiotic stress [100,101], being capable of increasing yield in crops such as potato [102]. This makes them an interesting carrier material [103]. It is not difficult to find some examples where this material is used as an agrochemical carrier for pesticides [104], herbicides [103], and fungicides [105]. Another interesting possibility is that, unlike alginate, chitosan can easily entrap oils [106] and hydrophobic compounds [107], which opens an easy possibility to work with these types of substances. Chitosan particles have also been used to improve plant fertilization [108], demonstrating how chitosan-enclosed fertilizers are capable of increasing maize crop yield [109,110]. Chitosan as an encapsulation material has been studied to include biostimulants, demonstrating how encapsulation utilization can be interesting to increase melatonin biostimulant properties against salinity [111]. In addition, using chitosan to perform as nanoparticles that enclose gibberellic acid is capable of improving tomato plant productivity by 77% [30]. In addition, chitosan nanoparticles were able to control the release of salicylic acid over a 7-day period, reducing the need for new treatments [29]. However, while research is promising, there is still a lack of research on biostimulant encapsulation with chitosan.

### 3.3. Carrageenan

Carrageenans (CGs) are hydrophilic polysaccharides found in numerous species of red macroalgae (Rhodophyta). Industrially, they are mainly extracted from macroalgal cells by a hot alkali extraction process [112]. Chemically, they are highly sulfated galactans, which give the polymers a strongly anionic character. These negatively charged polymers are linear and consist of repeating disaccharide units of D-galactose and 3,6-anhydro-D-galactose (3,6-AG) [113]. Carrageenan is divided into different types, such as λ, κ, ι, ε, and μ, all containing 22 to 35% sulfate groups (Figure 6). The main differences affecting the properties of the carrageenan type are the number and position of ester sulfate groups and the content of 3,6- AG. A higher content of ester sulfate means a lower solubility temperature and a lower gel strength. Kappa-type carrageenan has an ester sulfate content of about 25 to 30% and a 3,6- AG content of about 28 to 35%. Iota-type carrageenan has a sulfate ester content of about 28 to 30% and a 3,6- AG content of about 25 to 30%. Lambda-type carrageenan has a sulfate ester content of about 32 to 39% and no content of 3,6- AG Among the different types of CGs, ι and κ can form three-dimensional gels through interactions with certain metal ions such as potassium and calcium [114]. Carrageenan hydrogels are usually prepared in combination with other polymers. A detailed overview of their synthesis can be found in Zia et al. [115]. Carrageenan is an interesting option for encapsulation processes as it is a completely natural polymer and biodegradable [116].

Carrageenan is commonly used at the industrial level as a hydrogel to immobilize enzymes to improve stability, activity, and reusability [117]. In addition, carrageenan can be used to immobilize cells to run interesting biofactories, which can be used industrially to produce, for example, ethanol [118], or it can be used in fermentative processes [119]. Another interesting application with immobilized cells is the use of microalgae to remove nutrients from wastewater [120]. Other industrial applications can be found in the review by de Velde et al. [121]. Carrageenan hydrogels are being explored in agriculture to be added to soil, in order to increase fertilizers’ shelf life [122] through controlled release [123]. Another interesting application is about increasing the moisture content of the soil, as carrageenan is a super absorbent polymer that can improve water retention in the soil, which is important for plants during drought [124]. In addition, carrageenan is capable of improving growth in banana [125]. All this makes carrageenan an interesting polymer to explore as an encapsulation material for biostimulants. However, the agricultural applications beyond those mentioned above have not, as far as we know, been explored at the time of writing.

### 3.4. Guar Gum

Guar gum is a galactomannan obtained by grinding the seeds of *Cyamopsis tetratogonolobus (L.)* Taub., without pressing uronic acid, which distinguishes it from most plant gums [126]. In terms of its composition and structure, guar gum is mainly composed of high-molecular-weight polysaccharides of galactomannans, which are linear chains of (1→4)-linked β-D-mannopyranosyl units with (1→6)-linked α-D-galactopyranosyl residues as side chains [127] (Figure 7). The high number of branches in the guar gum structure may be responsible for its hydrating properties, as well as its greater hydrogen bonding activity [128]. It is capable of bonding with cellulosic materials and hydrated minerals such as kaolinite. It is also fully biodegradable, making it an interesting polymer for environmentally friendly encapsulations [129].

Guar gum can be used for microencapsulation and nanoencapsulation. The most common methods for drug delivery using the polymer are: (i) emulsions using glutaraldehyde as a cross-linking agent [130]; (ii) ionotropic gelation, which is a very simple and efficient method for preparing microspheres using safe reagents [131]. The most widely used method for formulating nanoparticles is the one- or two-step emulsion method, which yields a stable and uniform particle size depending on the type of drug [132]. Nowadays, guar gum is applied in the pharmacological, biomedical, and food industries [132]. Interestingly, some uses in agriculture have been explored, for example, using their production wastes as raw material for vermicompost [133,134]. In addition, the use of guar gum in agriculture is due to its superabsorbent properties, using it as a moisture-retaining material to prevent soil water loss [135]. In addition, it is demonstrated that this polymer can be used as a delivery system for micronutrients [136] or pesticides [137]. Guar gum explores biostimulant encapsulation, enclosing microbial biostimulants [138], and, in addition, can be easily associated with inorganic biostimulants such as silicon [139,140]. The superabsorbent properties of carrageenan and the ability to entrap active ingredients and control release, which have already been demonstrated [136,137], as well as the lack of research as a plant biostimulant as an encapsulating agent, make research under field conditions with guar gum particularly interesting for further study.

### 3.5. Xanthan Gum

Xanthan gum is isolated from *Xanthomonas* spp. Most of the production is obtained from *X. campestris* strains, but it can also be obtained from *X. citri subsp. citri, X. hortorum*, or *X. axonopodis* [141]. Xanthan gum is a heteropolysaccharide, with a primary structure that consists of a cellulose-like backbone of β-1,4-linked glucose units, substituted alternately with a trisaccharide side chain [142]. This side chain is composed of two mannose units separated by a glucuronic acid, where the internal mannose is mostly O-acetylated, and the terminal mannose may be substituted by a pyruvic acid residue (Figure 8) [141]. Xanthan gum solutions are highly viscous even at very low concentrations due to their strong polyelectrolyte nature. They show strong pseudoplastic behavior under shear. The viscosity depends on the polymer concentration, temperature, salt, and pH [141]. The polymer can be biodegraded into oligosaccharides, monosaccharides, and ultimately water and carbon dioxide [143], being an interesting biodegradable option for biostimulant encapsulation [144].

The synthesis of xanthan gum hydrogels is usually carried out by chemical or physical cross-linking methods in the presence of various natural and synthetic polymeric materials [145]. The polymer is of practical use due to its slow dissolution and strong swelling in biological fluids [146]. It can be used to produce nanoparticles with biocompatible components and environmentally friendly treatments [147]. Xanthan gum has a wide range of applications in biomedical engineering and agriculture [145]; in the latter, it is mainly used as a thickening agent for agrochemical formulations [141]. It has been used in the encapsulation of secondary metabolites [148], enzymes [149], and microorganisms [150] for controlled release. However, their use in agriculture remains scarcely explored [145], with a few studies limited to herbicides [151] and plant exudates for pest control [152] that demonstrated the polymer’s capacity to improve performance under field conditions. Xanthan gum was also explored for biochar encapsulation, which is an interesting way to enhance organic fertilization for better nutrient uptake [153]. Since xanthan gum is generally considered a non-gelling polymer, it tends to be underrepresented in studies investigating bio-based encapsulation materials for controlled-release systems in agricultural crops [141]. Despite its potential use as a biostimulant encapsulator, xanthan gum has yet to be widely explored for agricultural purposes [145]. For all these reasons, we believe that the use of xanthan gum as an encapsulating agent could be interesting to expand biostimulants properties under field conditions.

## 4. Bibliographic Study on Encapsulation in Agriculture

The information provided here demonstrates that encapsulation is an interesting technology for agriculture improvement. In addition, this field is becoming a hot topic, which can be easily shown in a bibliographic study, where 70% of the documents were created in the last five years (Figure 9), demonstrating the growing interest in applying encapsulation in agriculture.

After an association analysis (Figure 10), we found four different clusters: (i) the term agriculture, correlated with all other clusters, but was well linked to terms such as pesticide, herbicide, and fertilizer, and very strongly associated with the term nanoparticles; (ii) the second cluster, linked to the first by nanoparticles, had only two terms: Chitosan and Ionic Gelation; (iii) the third cluster is linked to the first by the term Agriculture and correlates Alginate with Carrageenan and Xanthan with Microcapsules and Emulsion Techniques, which are the only ones linked to Moisture Content; (iv) the last cluster is linked to Drug Delivery Systems, Electrospinning, and Hydrogel. Interestingly, the latter terms are used more recently in the bibliography, changed from fields such as biomedicine to agriculture. The timeline also illustrates that encapsulation in agriculture is increasingly shifting to nanoparticles rather than microcapsules and that chitosan is now the preferred polymer (Figure 10).

## 5. Cost–Benefits Analysis for the Use of Natural Polymers in Agriculture

As we have noted, chitosan is readily found in some agrochemical applications, such as pesticides [104], herbicides [103], fungicides [105], and even biostimulants [154]. However, most of the research conducted with chitosan uses a high degree of purity polymer, which is very expensive, but in the laboratory is the form that ensures a reasonable degree of deacetylation and homogeneity of molecular weight, which is important to ensure a repeatable encapsulation process [90]. For example, 50 g of low-molecular-weight chitosan (CAS Number 9012-76-4) costs EUR 121.0. Using 0.2% chitosan [155] to produce nanoparticles requires 2 g for 1 L of encapsulated biostimulants, resulting in an additional cost of EUR 5.0, which is too expensive for their use in agricultural formulations. The use of industrial chitosan might solve this price problem. However, at the time of writing this review, there is no known work comparing industrial-grade chitosan nanoparticles with laboratory-grade ones, which can allow a competitive price for nanoencapsulation in agriculture.

It is noteworthy that nanoparticle production is expensive [155], but microencapsulation is relatively cheap [156]. Sodium alginate (CAS Number 9005-38-3) at 1% is used to make microparticles [54], supposedly at a price five times lower than chitosan nanoparticles in 1 L of encapsulated biostimulants. Therefore, in addition to the simpler procedures to produce in the laboratory and to scale up, an interesting and profitable option for biostimulants encapsulation is expected.

## 6. Conclusions

Researchers need to explore the use of other polymers, such as those mentioned in this review, to improve the properties and durability of biostimulants under field conditions. Nowadays, xanthan gum is one of the most industrially relevant microbial polysaccharides with a competitive price [157]. Carrageenans and guar gum can serve as templates for agricultural industrial purposes with a low additional cost [121,158], providing not only control release as an encapsulation agent, but also interesting properties to increase the amount of moisture in the soil. The ones mentioned are scarcely studied and are an excellent choice for future encapsulation processes, which, in our opinion, can be an interesting option to be applied in agriculture production.

## Figures and Tables

**Figure 1 plants-12-00055-f001:**
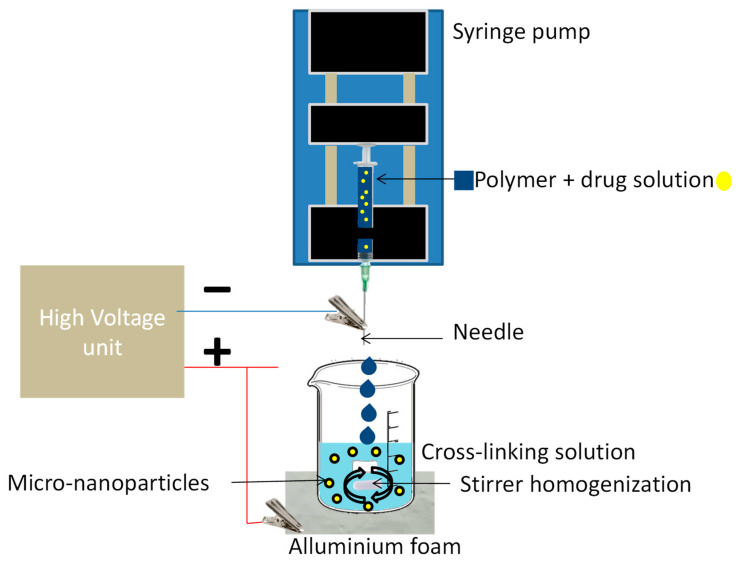
Encapsulation using electrospray techniques.

**Figure 2 plants-12-00055-f002:**
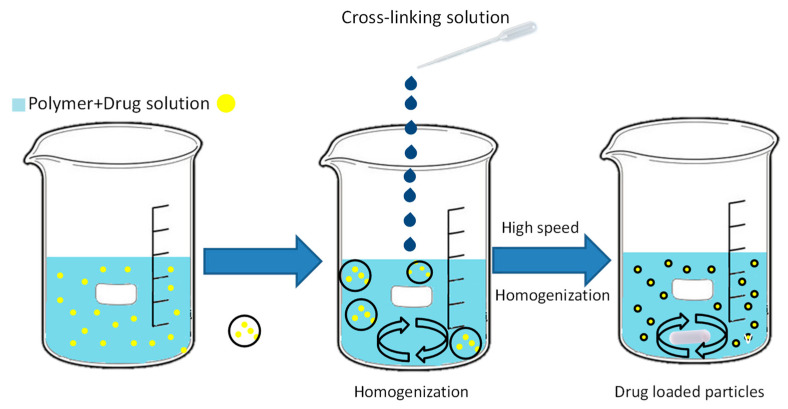
Encapsulation using ionotropic gelation techniques.

**Figure 3 plants-12-00055-f003:**
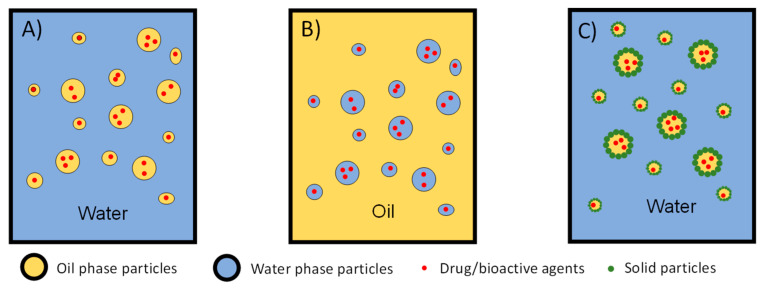
**Encapsulation using emulsion techniques.** (**A**) oil-in-water emulsion (O/W); (**B**) water-in-oil emulsion (W/O); (**C**) Pickering emulsion in an O/W.

**Figure 4 plants-12-00055-f004:**
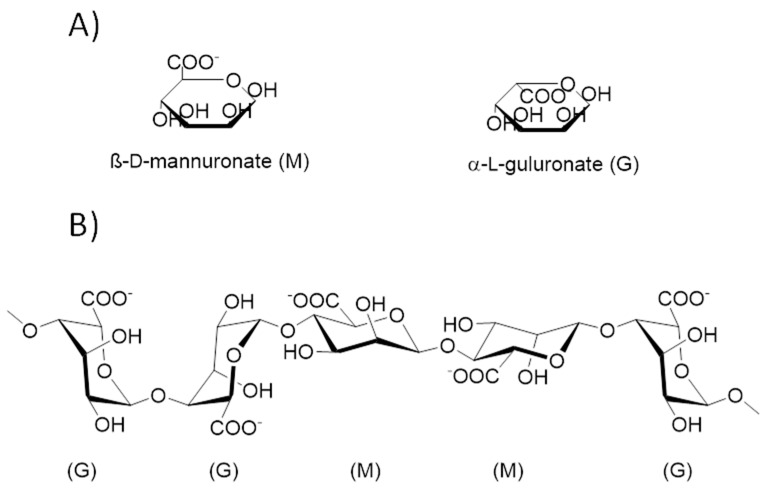
**Alginate structure:** (**A**) alginate monomer units; (**B**) alginate basic structure.

**Figure 5 plants-12-00055-f005:**
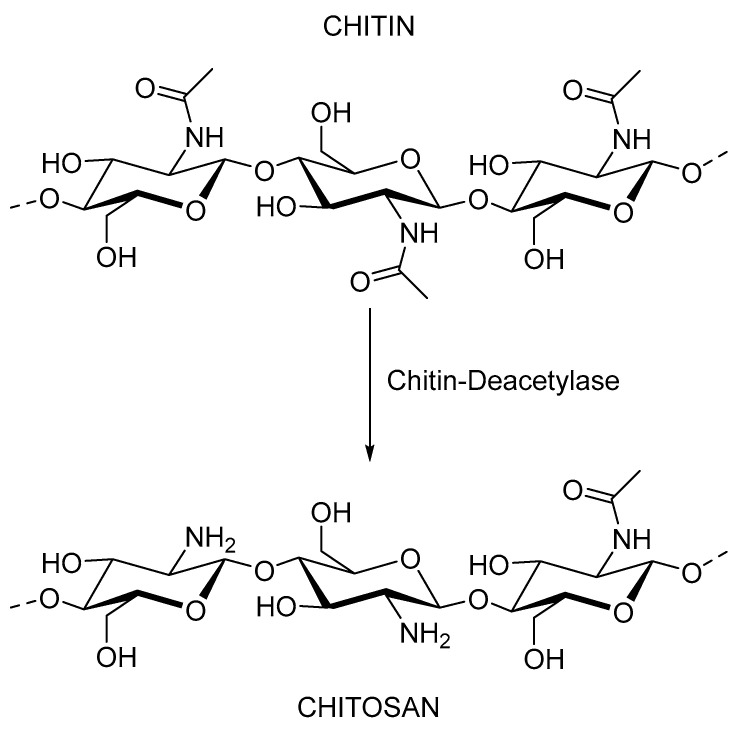
Chitin and chitosan structure.

**Figure 6 plants-12-00055-f006:**
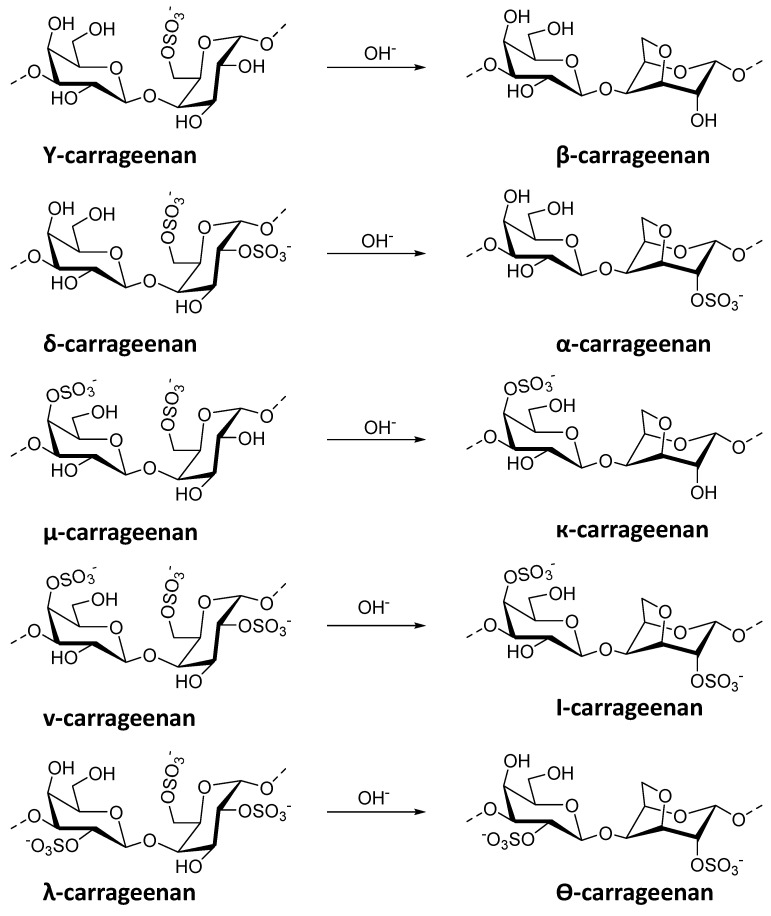
Main structures of carrageenans and derived carrageenans after alkaline hydrolysis.

**Figure 7 plants-12-00055-f007:**
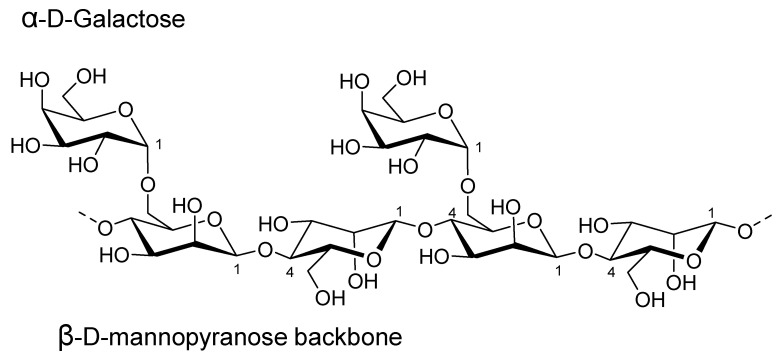
Guar gum structure.

**Figure 8 plants-12-00055-f008:**
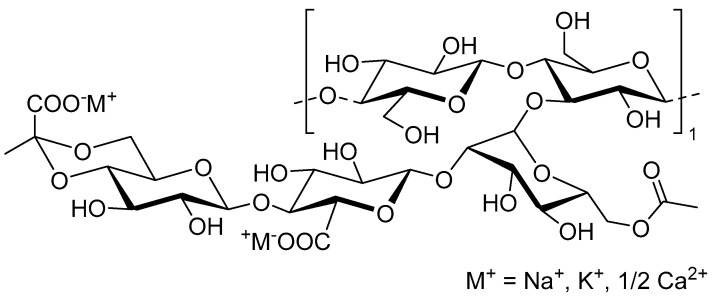
Xanthan gum structure.

**Figure 9 plants-12-00055-f009:**
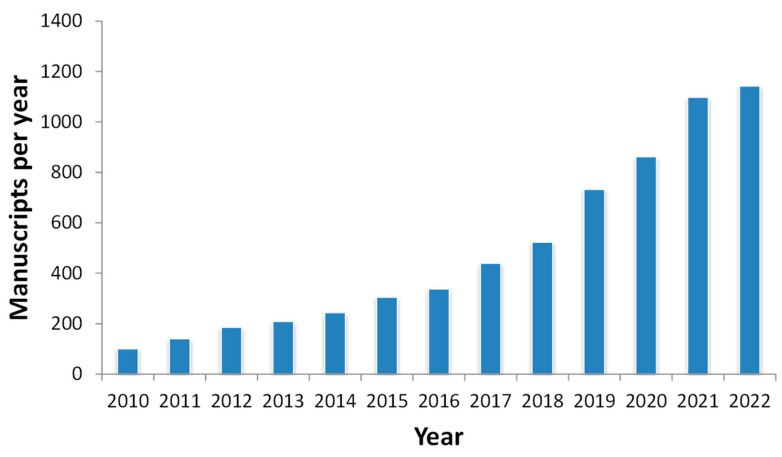
**Bibliographic citations since 2010.** The number of references was taken from WoS and Scopus. The keywords used in the study were agriculture plus encapsulation.

**Figure 10 plants-12-00055-f010:**
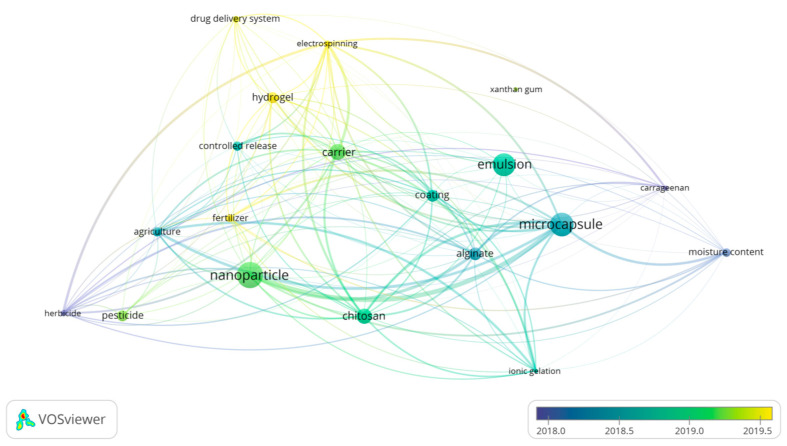
**Map of terms of the publications on encapsulation in agriculture.** The keywords “agriculture” and “encapsulation” were searched in the titles, abstracts, and keywords, during the period from 2010 to 2020. The analysis is based on all the publications that are classified as Articles. The distance between terms indicates how strong their relationship is. The color represents the age of the term, in the period between 2018 and 2020, where blue is closest to 2018, and yellow is close to 2020. The size indicates the number of publications in which the term appears. References were taken from WoS and Scopus.

## Data Availability

Not applicable.

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
