# Peer review of "Encapsulation with Natural Polymers to Improve the Properties of Biostimulants in Agriculture"

_plants, 2022, doi:10.3390/plants12010055_

Round 1

Reviewer 1 Report

Dear authors,

this review paper gives an overview of encapsulation methods, and the paper itself is generally written with only a hint about biostimulators. So, first of all, I think the title is inappropriate; it would better be „An overview of different encapsulation techniques with natural polymers – new possibilities for agriculture. “ Also, the Abstract needs to be adjusted to the title; no emphasis should be placed on biostimulators. Also, there are several technical issues throughout the whole text; here they are:

-          There is a number of words written with a dash  (L38, L77, L84, L85, L391)

-          L 94 there is no dot after et al

-          L95 CaCO3, 3 should be in superscript

-          L 97 there are two dots after Fig. 1 title

-          Please check the name of species throughout the whole text; they should be in Italics (L 142, L293,  L322, L323, L324)

-          L157, there is no dot at the end of a sentence

-          L162 M and G abbreviations are not described

-          L163 dot should be after brackets

-          L187 Al3+,3+ should be in superscript

-          L246 what is TPP

-          L257-269 rewrite the sentence; it is not comprehensible

-          In L268, you write about six basic CGs forms, and in Fig. 6 are ten different forms; what about that?

-          L271can be found in…what ?

-          L298-302 rewrite the sentence; it is not comprehensible

-          in each chapter, emphasis should be placed on the application of different natural polymers in agriculture and extend the text about the benefits of this application on yield or quality of agricultural products

-          the conclusion is all about cost-benefit analysis and analysis of the bibliographic study of encapsulation. In my opinion, that should be in the last chapter, and the conclusion should give future remarks on this subject.

If all the suggestions are considered, and appropriate changes are made, the paper should be considered for publishing in Plants.

Author Response

Dear Reviewer, Thank you for your comments and suggestions. We have decided to change the title after reading the comments of the two reviewers, the new one is "Encapsulation with natural polymers to improve the properties of biostimulants in agriculture".

Our idea, with this article, is to introduce researchers working with biostimulants to the possibilities of encapsulation to improve the properties and shelf life of these types of compounds. Why only biostimulants? For two main reasons:

1: we believe that we can easily find interesting and update reviews on the encapsulation of fertilizers (e.g. https://doi.org/10.1515/revce-2020-0044; https://doi.org/10.1016/j.jclepro.2021.127018; https://doi.org/10.1080/01904167.2020.1744647) or agrochemicals (e.g.: DOI: 10.1039/D0EM00404A; https://doi.org/10.1021/acs.jafc.9b06982; https://doi.org/10.1016/j.cis.2022.102645).

2: encapsulation of biostimulants is hardly studied, but in our opinion, it is an interesting way to overcome the weaknesses of biostimulants and improve their positive properties. This review is intended to be the first step for future researchers in this field by providing some protocols and ideas to get started.

I hope you find our explications appropriate. We made changes following your suggestions

          There is a number of words written with a dash  (L38, L77, L84, L85, L391)

Corrected

-          L 94 there is no dot after et al

Dot added

-          L95 CaCO3, 3 should be in superscript

Done

-          L 97 there are two dots after Fig. 1 title

Erased

-          Please check the name of species throughout the whole text; they should be in Italics (L 142, L293,  L322, L323, L324)

Done

-          L157, there is no dot at the end of a sentence

Done

-          L162 M and G abbreviations are not described

Thanks for the apreciation I described it now as:

β-(1→4) linked D-mannuronic acid residues (M) and α -(1→4) linked L-guluronic acid residues (G) residues

-          L163 dot should be after brackets

Done

-          L187 Al3+,3+ should be in superscript

Done

-          L246 what is TPP

Thanks for the annotion we change by the name without abbreviation „tripolyphosphate”

-          L257-269 rewrite the sentence; it is not comprehensible

We change both paragrap.

The paragraph that belongs to chitosan chapter has been changed by (L261 to 276)

Chitosan are widely studied as plant biostimulants  because of their properties against abiotic stress [100,101], being capable to increase yield in crops as potato [102]. This makes them an interesting carrier material [103]. It is not difficult to find some exam-ples where this material is used as an agrochemical carrier for pesticides [104], herbicides [103] and fungicides [105]. Another interesting possibility is that, unlike alginate, chitosan can easily entrap oils [106] and hydrophobic compounds [107] which opens an easy pos-sibility to work with these types of substances. Chitosan particles have also been used to improve plant fertilization [108], demonstrating how chitosan enclosed fertilizers are ca-pable to increase maize crop yield [109,110]. Chitosan as an encapsulation material has been studied to include biostimulants, demonstrating how encapsulation utilization can be interesting to increase melatonin biostimulants properties against salinity [111]. In ad-dition, using chitosan to perform nanoparticles that encloses gibberellic acid, are capable to improved Tomatoes plant productivity by 77% [30]. In addition, chitosan nanoparticles were able to control the release of salicylic acid over a 7-day period, reducing the need for new treatments [29]. However, while the research is promising, there is still a lack of re-search in biostimulant encapsulation with chitosan.

An the paragraph that belongs to carragenans is changed by

L278 to 296

Carrageenans (CGs) are hydrophilic polysaccharides found in numerous species of red macroalgae (Rhodophyta). Industrially, they are mainly extracted from macroalgal cells by a hot alkali extraction process [112]. Chemically, they are highly sulphated galac-tans, which give the polymers a strongly anionic character. These negatively charged pol-ymers are linear and consist of repeating disaccharide units of D-galactose and 3,6-anhydro-D-galactose (3,6- AG ) [113]. Carrageenan is divided into different types such as λ, κ, ι, ε, μ, all containing 22 to 35% sulphate groups (Figure 6). The main differences af-fecting the properties of the carrageenan type are the number and position of ester sul-phate groups and the content of 3,6- AG. A higher content of ester sulphate means a lower solubility temperature and a lower gel strength. Kappa-type carrageenan has an ester sul-phate content of about 25 to 30% and a 3,6- AG content of about 28 to 35%. Iota-type carra-geenan has a sulphate ester content of about 28 to 30% and a 3,6- AG content of about 25 to 30%. Lambda-type carrageenan has a sulphate ester content of about 32 to 39% and no content of 3,6- AG  Among the different types of CGs, ι and κ can form three-dimensional gels through interactions with certain metal ions as potassium and calcium [114]. Carra-geenan hydrogels are usually prepared in combination with other polymers. A detailed overview of their synthesis can be found in Zia et al. [115]. Carrageenan is an interesting option for encapsulation processes as it is a completely natural polymer and biodegrada-ble [116].-         

 In L268, you write about six basic CGs forms, and in Fig. 6 are ten different forms; what about that?

Carragenans have five basic forms (now is corrected in the text) and the other proceds from this by alcali hidrolysis to point that we change the Figure 6 title as: Main structures of carrageenans and derived carrageenans after alkaline hydrolysis.

-          L271can be found in…what ?

The sentance has changed by:

Other industrial applications can be found in the review by de Velde et al.  [121].”

-          L298-302 rewrite the sentence; it is not comprehensible

The sentence is rewrite as:

The high number of branches in the guar gum structure may be responsible for its hydrat-ing properties as well as its greater hydrogen bonding activity [128].  It´s capable to bond with cellulosic materials and hydrated minerals such as kaolinite. It is also fully biode-gradable, making it an interesting polymer for environmentally friendly encapsulations [129].

-          in each chapter, emphasis should be placed on the application of different natural polymers in agriculture and extend the text about the benefits of this application on yield or quality of agricultural products

At the end of polymers chapters the paragraphs was changed  to emphatise the polymer utilization in agriculture:

3.1. Alginate

Encapsulation with alginate is used in some agrochemical formulations, especially to control the release of active ingredients [63], for example, to control the release of pesti-cides [64,65], herbicides [66] or fungicides [67]. Another use is the optimization of the ferti-lization process by encapsulating various fertilizers such as biochar [68] or biofertilisers [69]. In the agricultural industry, alginate is also used as a superabsorbent polymer to coat seeds, fruits and vegetables and as a carrier of bacteria and fungi to promote plant growth and biocontrol [70]. Alginate hydrogel combined with some amino acids has been used as coating agent, to be used in seeds for commercial applications, to stimulates the early growth of plants and thus leading to higher crop yields [71].  Alginate beads have been used to improve the properties of biostimulants, especially in microorganism-based for-mulations [72]. Encapsulation within alginic matrixes attempts to increase the viability of cells and use them for various purposes, such as increasing the tolerance of plants to drought [73–75] or salinity [76–78] and the uptake of nutrients [79]. Encapsulation of mi-croorganism- based biostimulants with alginate is a promising way to extend their shelf life [80]. Alginate beads can improve the properties of biostimulants, are capable to in-crease plant growth of lettuce [81], and increase wheat yield under drought conditions by entrapping N2-fixing bacteria [82]. However, at the time of writing, the properties resulting from encapsulation have hardly been researched under field conditions, as there are no corresponding production trials. In addition, we could not find any use of alginate for en-capsulating other types of biostimulants as pure or organic agents against abiotic stress. Alginate in our opinion can be an interesting and profitable way to improve plant bi-ostimulant dosage in field conditions as is demonstrated in other interesting agrochemi-cals products [83], however further research is needed.

3.2. Chitosan

Chitosan are widely studied as plant biostimulants because of their properties against abiotic stress [100,101], being capable to increase yield in crops as potato [102]. This makes them an interesting carrier material [103]. It is not difficult to find some exam-ples where this material is used as an agrochemical carrier for pesticides [104], herbicides [103] and fungicides [105]. Another interesting possibility is that, unlike alginate, chitosan can easily entrap oils [106] and hydrophobic compounds [107] which opens an easy pos-sibility to work with these types of substances. Chitosan particles have also been used to improve plant fertilization [108], demonstrating how chitosan enclosed fertilizers are ca-pable to increase maize crop yield [109,110]. Chitosan as an encapsulation material has been studied to include biostimulants, demonstrating how encapsulation utilization can be interesting to increase melatonin biostimulants properties against salinity [111]. In ad-dition, using chitosan to perform nanoparticles that encloses gibberellic acid, are capable to improved tomatoes plant productivity by 77% [30]. In addition, chitosan nanoparticles were able to control the release of salicylic acid over a 7-day period, reducing the need for new treatments [29]. However, while the research is promising, there is still a lack of re-search in biostimulant encapsulation with chitosan.

3.3. Carrageenan

We have found only one other manuscript on the subject of  carrageenan agricultural use, which we have inserted in [125] and rephrased the last paragraph as follows:

Carrageenan is commonly used at the industrial level as a hydrogel to immobilize enzymes to improve stability, activity, and reusability [117]. In addition, carrageenan can be used to immobilize cells to run interesting biofactories, that can be used industrially to produce, for example, ethanol [118], or it can be used in fermentative processes [119]. An-other interesting application with immobilized cells is the use of microalgae to remove nu-trients from wastewater [120]. Other industrial applications can be found in the review by de Velde et al.  [121]. Carrageenan hydrogels are being explored in agriculture to be added to soil, in order to increase fertilizers' shelf life [122] through controlled release [123]. An-other interesting application is about increasing the moisture content of the soil, as carra-geenan is a super absorbent polymer that can improve water retention in the soil, which is important for plants during drought  [124]. In addition, carrageenan is capable to im-proves growth in banana [125]. All this makes carrageenan an interesting polymer to ex-plore as an encapsulation material for biostimulants. However, the agricultural applica-tions beyond the mentioned above, have not, as far as we know, been explored at the time of writing.

3.4. Guar gum

We cannot find any other relevant studies using guar gum as an encapsulation system in agriculture, only some on its use as a coating agent to improve the shelf life of fruit, but we think this is outside the scope of the manuscript. We add the following sentence to indicate the interest in its use as an encapsulant in agriculture:

The superabsorbent properties of carrageenan and the ability to entrap active ingredients and control release, which have already been demonstrated [136,137], as well as the lack of research as a plant biostimulant as an encapsulating agent, make research under field conditions with guar gum particularly interesting for further study.

3.5. Xhantan gum

As the guar gum, Xanthan gum is scarcely studied in agricultural applications, however some interesting reports point out the utility as natural encapsulation agent in agriculture. We add the following sentence to indicate the interest in its use as an encapsulant in agriculture:

Since xanthan gum is generally considered a non-gelling polymer, it tends to be un-derrepresented in studies investigating bio-based encapsulation materials for control re-lease system in agricultural crops [141]. Despite its potential use as a biostimulant encap-sulator, xantham gum has yet to be widely explored for agricultural purposes [145]. For all these reasons, we believe that the use of xanthan gum as an encapsulating agent could be interesting to expand biostimulants properties under field conditions.

-          the conclusion is all about cost-benefit analysis and analysis of the bibliographic study of encapsulation. In my opinion, that should be in the last chapter, and the conclusion should give future remarks on this subject.

We perform separete bibliographic study and cost benefit analysis, and added concluisons that covers future remarks

  1. Bibliographic study on encapsulation in agriculture

The information provided here demonstrates that encapsulation is an interesting technol-ogy for agriculture improvement. In addition, this field is becoming a hot topic, which can be easily shown in a bibliographic study, where 70% of the documents were done in the last five years (Figure 9), demonstrating the growing interest in applying encapsulation in agriculture.

After an association analysis (Figure 10), we found four different clusters: i) the term agriculture, correlated with all other clusters, but was well linked to terms such as pesti-cide, herbicide, and fertilizer, and very strongly associated with the term nanoparticles; ii) the second cluster, linked to the first by nanoparticles, had only two terms: Chitosan and Ionic Gelation; iii) the third cluster is linked to the first by the term Agriculture and corre-lates Alginate with Carrageenan and Xanthan with Microcapsules and Emulsion Tech-niques, which are the only ones linked to Moisture Content; iv) the last cluster is linked to Drug Delivery Systems, Electrospinning, and Hydrogel. Interestingly, the latter terms are used more recently in the bibliography, changed from fields such as biomedicine to agri-culture. The timeline also illustrates that encapsulation in agriculture is increasingly shift-ing to nanoparticles rather than microcapsules and that chitosan is now the preferred polymer (Figure 10).

  1. Cost-benefits analysis for the use of natural polymers in agriculture

As we have noted, chitosan is readily found in some agrochemical applications such as pesticides [104], herbicides [103] and fungicides [105], and even biostimulants [154]. However, most of the research conducted with chitosan uses a high degree of purity polymer, which is very expensive, but in the laboratory is the form that ensures a rea-sonable degree of deacetylation and homogeneity of molecular weight, which is important to ensure a repeatable encapsulation process [90]. For example, 50 g of low molecular weight chitosan (CAS number 9012-76-4) cost 121.0 €. Using 0.2% chitosan [155], to produce nanoparticles requires 2 grams for 1 liter of encapsulated biostimulants, resulting in an additional cost of 5.0 €, which is too expensive for their use in agricultural formulations. The use of industrial chitosan might solve this price problem. However, at the time of writing this review, there is no known work comparing industrial-grade chitosan nanoparticles with laboratory-grade ones, such can allow a competitive price for nanoencapsulation in agriculture.

It's noteworthy, that nanoparticle production is expensive [155], but microencapsulation is relatively cheap [156]. Sodium alginate (CAS number 9005-38-3) at 1% is used to make microparticles [54] suppose a price five times lower than chitosan nanoparticles in 1 liter of encapsulated biostimulants. That in addition to the simpler procedures to produce in the laboratory and to scale up, an interesting and profitable option for biostimulants encapsulation is expected.

  1. Conclusion

Researchers need to explore the use of other polymers, such as those mentioned in this review, to improve the properties and durability of biostimulants under field conditions. Nowadays, xanthan gum is one of the most industrially relevant microbial polysaccharides with competitive price [157]. Carrageenan’s and guar gum can serve as templates for agricultural industrial purposes with a low additional cost [121,158], providing, not only control release as encapsulation agent, but also its interesting properties to increase the amount of moisture in the soil. The ones mentioned are scarcely studied and are an excellent choice for future encapsulation processes, that in our opinion can be an interesting option to be applied in agriculture production

Reviewer 2 Report

This is a very useful paper and has a potential to be cited in other papers. I have only a few small comments:

There is no necessity to say in title that this is a beginner’s guide. The title should be as short as possible and underline the important information.

There are several interpunction mistakes in the text and the figure description.

There is something wrong with Figure 10, at least in my copy – it is hardly visible on the left margin of the text. 

Author Response

This is a very useful paper and has a potential to be cited in other papers. I have only a few small comments:

Thanks for you comments and it´s a pleasure that you enjoy reading our paper

There is no necessity to say in title that this is a beginner’s guide. The title should be as short as possible and underline the important information.

As your suggestions we change our title as:

Encapsulation with natural polymers to improve the properties of biostimulants in agriculture

There are several interpunction mistakes in the text and the figure description.

Thanks for your annotation, we  fix it

There is something wrong with Figure 10, at least in my copy – it is hardly visible on the left margin of the text. 

I think is a problem with word document, we can provide a full image if it necessary.

Round 2

Reviewer 1 Report

Dear authors,

thank you for all corrections in the text. In my opinion, the paper looks much better and I think that it should be published in Plants in its present form.